# Dose-Dependent Effect of Hyperbaric Oxygen Treatment on Burn-Induced Neuropathic Pain in Rats

**DOI:** 10.3390/ijms20081951

**Published:** 2019-04-20

**Authors:** Zong-Sheng Wu, Sheng-Hua Wu, Su-Shin Lee, Cen-Hung Lin, Chih-Hau Chang, Jing-Jou Lo, Chee-Yin Chai, Ching-Shuang Wu, Shu-Hung Huang

**Affiliations:** 1Department of Medical Laboratory Science and Biotechnology, College of Health Sciences, Kaohsiung Medical University, Kaohsiung 807, Taiwan; a8905114@gmail.com (Z.-S.W.); m785034@kmu.edu.tw (C.-S.W.); 2Department of Anesthesiology, Kaohsiung Medical University Hospital, Kaohsiung 807, Taiwan; elsawu2@gmail.com; 3Department of Anesthesiology, Kaohsiung Municipal Hsiao-Kang Hospital, Kaohsiung 807, Taiwan; 4Department of Anesthesiology, School of Medicine, College of Medicine, Kaohsiung Medical University, Kaohsiung 807, Taiwan; 5Division of Plastic Surgery, Department of Surgery, Kaohsiung Medical University Hospital, Kaohsiung 807, Taiwan; sushin@kmu.edu.tw (S.-S.L.); igor8301023@yahoo.com.tw (C.-H.C.); 6Department of Surgery, School of Medicine, College of Medicine, Kaohsiung Medical University, Kaohsiung 807, Taiwan; 7Center for Stem Cell Research, Kaohsiung Medical University, Kaohsiung 807, Taiwan; 8Orthopaedic Research Center, Kaohsiung Medical University, Kaohsiung 807, Taiwan; 9Department of Plastic & Reconstructive Surgery, Kaohsiung Chang Gung Memorial Hospital, Chang Gung University College of Medicine, Kaohsiung 807, Taiwan; gigilin119@msn.com; 10School of Post-Baccalaureate Medicine, College of Medicine, Kaohsiung Medical University, Kaohsiung 807, Taiwan; joekll@hotmail.com; 11Departments of Pathology, Kaohsiung Medical University Hospital, Kaohsiung Medical University, Kaohsiung 807, Taiwan; cychai@kmu.edu.tw

**Keywords:** melatonin, opioid receptor, neuropathic pain, hyperbaric oxygen, cuneate nucleus

## Abstract

Hyperbaric oxygen treatment (HBOT) has been used to reduce neuropathic pain. Melatonin and opioid receptors are involved in neuropathic pain, but it is not known if HBOT works through these pathways to achieve its antinociceptive effect. We divided anesthetized rats into two treatment and three sham groups. The two treatment groups received third-degree burns on their right hind paws, one treated in a hyperbaric chamber for a week and the other for two weeks. We evaluated the mechanical paw-withdrawal threshold (MWT) and expression of melatonin receptor 1 (MT1), melatonin receptor 2 (MT2), μ (MOR) and κ (KOR) opioid receptor, brain-derived neurotrophic factor (BDNF), Substance P, and calcitonin gene-related peptide (CGRP) in cuneate nucleus, dorsal horn, and hind paw skin by immunohistochemical, immunofluorescence assays and real-time quantitative polymerase chain reaction (RT-PCR). The group receiving one-week HBOT had increased expressions of MT1, MT2, MOR and KOR and decreased expressions of BDNF, Substance P, and CGRP. Their mechanically measured pain levels returned to normal within a week and lasted three weeks. This anti-allodynia effect lasted twice as long in those treated for two weeks. Our findings suggest that increasing the duration of HBOT can reduce burn-induced mechanical allodynia for an extended period of time in rats. The upregulation of melatonin and opioid receptors observed after one week of HBOT suggests they may be partly involved in attenuation of the mechanical allodynia. Downregulation of BDNF, substance P and CGRP may have also contributed to the overall beneficial effect of HBOT.

## 1. Introduction

Millions of people worldwide suffer from burn-induced neuropathic pain, a chronic pain state induced by burn-induced lesions or injury of the peripheral or central nervous system [1]. In neuropathic pain, there is allodynia, hyperalgesia, and spontaneous pain, which can persist from months to years [2]. No effective analgesic or other treatments have been found to adequately manage neuropathic pain. Such continuous pain causes great physical and emotional discomfort, lowering the quality of life and requires many medical resources to manage [3].

In hyperbaric oxygen treatment (HBOT), patients are treated with 100% oxygen under pressures higher than 1.4 absolute atmospheres (ATA), usually to promote wound healing, dilute carbon monoxide poisoning, and reduce inflammatory reactions [4,5,6]. Recently, two different studies, both in models of chronic constriction injury (CCI), have reported that HBOT effectively reduced neuropathic pain [7,8]. HBOT has also been used in the treatment of neurological diseases and the reduction of chronic pain [9,10,11]. We previously found that HBOT could alleviate mechanical allodynia in a burn-induced neuroinflammation rat model [12]. At that time, we wondered whether an additional benefit could be achieved by extending the duration of HBOT and which pathways might be involved in HBOT’s effect on burn-induced neuropathic pain.

Melatonin (*N*-acetyl-5-methoxytryptamine) is secreted at night by the pineal gland in the brain [13]. This hormone may potentially be beneficial in the treatment of several diseases by virtue of its antioxidant properties, regulation of the circadian rhythms, and its antinociceptive effect under normal physiological conditions [14,15]. It has been found to have analgesic effects in the treatment of chronic pain, including abdominal pain, fibromyalgia, and neuropathic pain [13,16,17,18,19]. Melatonin receptors are found in various brain areas, retina, the pituitary gland and some peripheral organs [20,21,22,23]. Melatonin receptor 1 (MT1) and melatonin receptor 2 (MT2) play important roles in anti-nociceptive actions in the brain and spinal cord [18,19,24,25,26].

Opioids have been used for centuries to treat pain. Opioid receptors can be found in the brain, spinal cord, and central and peripheral neurons [27]. The main opioid receptors, μ (MOR) and κ (KOR), are involved in neuroprotection, nociception and inflammation [28] and their activation produces significantly higher analgesic effects [29]. Moreover, these opioid receptors are increased in sites of neural injury and have been found to have greater activity in rat models of chronic constriction injury (CCI) [30,31,32]. It is not known whether the antinociceptive effects of HBOT come about via the melatonin and opioid pathways.

In the present study, we used a burn-injury-induced neuropathic pain rat model to investigate the dose-dependent effect of HBOT on neuropathic pain by measuring its effects on mechanical pain and the expression of MT1, MT2 and opioid receptors, as well as markers of pain in the area of injury and two known pain transmission pathways, the dorsal horn, and cuneate nucleus [33,34].

## 2. Results

### 2.1. HBOT Dose-Dependent Effect on Mechanical Allodynia

As can be seen in Figure 1a,b, at four weeks post-burn, the burned rats that were not to receive HBOT and those that were to receive the therapy had the same baseline mechanical withdrawal thresholds (32.9 ± 0.58 g mechanical paw-withdrawal threshold (MWT)). One week after HBOT was initiated, the MWT of burned rats that received HBOT reached normal levels (47.4 ± 3.1 g MWT) and remained normal for three weeks (48.2 ± 2.1 g MWT). However, those levels then dropped again (33.21 ± 0.43 g MWT). One treatment group received HBOT for a week, and the other group continued to receive HBOT for another week. After two weeks, rats that continued to receive HBOT maintained normal MWT (46.03 ± 0.8 g MWT) for at least six weeks, the end of the 13-week study, see Figure 1a,b. These results suggested that HBOT had a dose-dependent effect, the more sessions of HBOT, the longer the anti-allodynia effect.

### 2.2. The Effect of HBOT on Expressions of MT1, MT2, MOR and KOR in the Dorsal Horn and Cuneate Nucleus

We performed immunohistochemical, immunofluorescence and Real-Time Quantitative Polymerase Chain Reaction (RT-PCR) studies of the dorsal horn of the spinal cord and cuneate nucleus of the medulla oblongata to assess the effect of HBOT on this pain transmission pathway. The results of the immunohistochemical studies showed that burned rats that had received one week of HBOT had significantly higher expressions of the melatonin and opioid receptors (MT1, MT2, MOR and KOR) than the untreated rats in both the cuneate nucleus (MT1: 7% ± 1% vs. 0.4% ± 0.05%, MT2: 13% ± 1.15% vs. 0.3% ± 0.05%, MOR: 5% ± 0.5% s. 0.3% ± 0.05% and KOR: 9.7% ± 0.85% vs. 0.1% ± 0.01%, *p* < 0.001) and dorsal horn (MT1: 12% ± 1.15% vs. 0.4% ± 0.05%, MT2: 8% ± 1.15% vs. 0.26% ± 0.05%, MOR: 9% ± 0.5% vs. 3% ± 1.15%, and KOR: 7% ± 1% vs. 2% ± 0.5%, *p* < 0.001), see Figure 2a,b and Figure 3a,b. The expressions of MT1, MT2, MOR and KOR were not upregulated after two weeks of HBOT in the dorsal horn of the spinal cord or the cuneate nucleus of the medulla oblongata. There were no significant differences between the burned rats receiving two weeks of HBOT and the burned rats that did not. We performed RT-PCR on medulla tissues only. The results of our RT-PCR showed that burned rats that had received one week of HBOT had significantly higher expressions of the melatonin receptors (MT1, MT2) than the burned that did not receive HBOT (MT1: 9.7 ± 0.5 vs. 1.5 ± 0.5, MT2: 6.9 ± 0.5 vs. 1.6 ± 0.5, *p* < 0.001), see Figure 3c. Together, these results suggest that HBOT increased the expressions of MT1, MT2, MOR and KOR in this pain transmission pathway, resulting in the relief of neuropathic pain. We performed immunofluorescence studies on dorsal horn tissues only. The results of these studies revealed that the burned rats that had received one week of HBOT had a significantly increased expression of MT1 in NeuN positive cells and MT2 in NeuN positive cells in the dorsal horn, compared to the burned rats that had not received HBOT (MT1 in NeuN positive cells: 30% ± 1.15% vs. 1% ± 0.5%, MT2 in NeuN positive cells: 12% ± 1% vs. 1% ± 0.5%, *p* < 0.001), see Figure 4.

### 2.3. HBOT Suppression of Astrocyte Activation in the Dorsal Horn

To understand the relationship between astrocytes and burn-induced neuropathic pain, we measured the expression of glial fibrillary acidic protein (GFAP), a marker of astrocytes, in the dorsal horn in HBOT-treated and untreated burned rats. After one week, the rats treated with HBOT for a week had significantly lower expressions of GFAP than the untreated burned rats (14% ± 0.9% vs. 45% ± 0.5%, *p* < 0.001). After two weeks, the burned rats treated with HBOT for two weeks were found to have lower expressions of GFAP than those treated for one week (5% ± 1.15% vs. 14% ± 0.9%, *p* 0.056), see Figure 5. There was no significant difference between burned rats treated with two weeks of HBOT and the burned rats that were not treated.

### 2.4. HBOT Alleviation of Nociceptive Reaction in the Dorsal Horns and Right Hind Paw Skin

We performed immunohistochemical studies to quantify the expressions of the pain-related neuropeptides brain-derived neurotrophic factor (BDNF) in the dorsal horn as well as Substance P, and calcitonin gene-related peptide (CGRP) in paw skin, both in treated and untreated burned rats. As can be seen in Figure 6, although BDNF was significantly lower in the dorsal horn of treated burned rats than the untreated ones, this effect was not found to have a dose-dependent effect in our comparison of the one- and two-week HBOT treatment groups (treated rats vs. untreated rats, 2% ± 0.2% vs. 28% ± 0.5% of one-week HBOT, 5% ± 0.5% vs. 37% ± 1% of two-week HBOT, *p* < 0.001). Figure 7 shows that the expressions of Substance P and CGRP in paw skin of the rats treated with HBOT were significantly lower than they were in the paws of the untreated rats and that these expressions were even lower in those receiving HBOT for two weeks (Substance P: 20% ± 1.15% vs. 38% ± 1% of one-week HBOT, 1% ± 0.11% vs. 22% ± 1.15% of two-week HBOT; CGRP: 8% ± 0.5% vs. 23% ± 0.9% of one-week HBOT, 5% ± 0.5% vs. 22% ± 1.15% of two-week HBOT), see Figure 7. These findings indicate that HBOT had a dose-dependent anti-allodynia effect in hind paw skin.

## 3. Discussion

HBOT had a dose-dependent effect on burn-induced neuropathic pain in rats in this study. It increased the mechanical pain threshold and increased the melatonin and opioid receptors in burned rats treated with HBOT for one week and decreased pain-related neuropeptides in tissue sections. The expressions of MT1, MT2, MOR and KOR were not upregulated in burned rats treated with HBOT for two weeks. These findings suggest that HBOT probably alleviated burn-induced neuropathic pain by increasing melatonin receptors and opioid receptors in the first week of HBOT and downregulating pain-related neuropeptides in the brain-to-site-of-injury pathway, as depicted in Figure 8. The analgesic effect of one week of HBOT was only maintained for three weeks, while the analgesic effect of two weeks HBOT was maintained for at least six weeks, until the end of the study. Also, at the end of the study, we found lower expressions of the pain-related neuropeptides (Substance P and CGRP) in the hind paw skin of the burned rats treated with HBOT for two weeks than in those of burned rats treated for one week only.

This study, using a burn-induced model of neuropathic pain, found that HBOT had a dose-dependent anti-allodynia effect. Other studies on HBOT have reported similar findings [35,36]. In one recent study on the antinociceptive effect of HBOT in mice, HBOT was found to have a dose-dependent opioid-mediated effect on the duration of pain reduction [37]. Another study, the results of which were similar to ours, found patients with lower limb complex regional pain syndrome treated for three weeks had less pain, swelling, and allodynia than those receiving one week of treatment [31]. Our behavioral test showed that the antinociceptive effect of two weeks of HBOT lasted longer than that of one week of HBOT. Two known pain-related neuropeptides, Substance P and CGRP, have been associated with many pain syndromes, including knee joint pain and lower back pain [38,39]. Our immunohistochemical studies of the right hind paw skin showed lower expressions of Substance P and CGRP in the rats receiving HBOT for two weeks than those receiving HBOT for one week only.

Melatonin has been found to have an analgesic effect on the spinal nociception and to decrease neuroinflammation [40,41]. In an animal model study in which pain was induced by ligating the sciatic nerve, the injection of melatonin led to an increase of the mechanical pain threshold and a decrease in thermal hyperalgesia [42]. Melatonin has been found to regulate pain perception via its receptors MT1/MT2 [43]. MT1 and MT2 can be found in the brain, thalamus, and dorsal horns of the spinal cord [44]. These receptors have been found to be involved in melatonin’s anti-nociceptive effect in inflammatory pain as well as neuropathic pain [45], the latter is achieved by moderating pain transmitted downwardly from the brain [46]. MT2 also mediates the inhibition of cyclic adenosine monophosphate (AMP) accumulation and the downregulation of intracellular Ca^2+^, diacylglycerol and arachidonic acid [47,48]. Melatonin’s antiallodynic effect, which it produces via increasing the amount of β-endorphin, has been found to be achieved through the activation of MT2 and opioid receptors [25].

Both melatonin receptors and opioid receptors have been positively associated with neuropathic pain and tactile allodynia [25,49]. In one opioid antagonist study, the therapeutic effects of HBOT on neuropathic pain were found to be suppressed when the opioid receptors were blocked [50]. In addition, HBOT has not been found to adequately relieve pain in opioid-tolerant mice, in which the two opioid receptors MOR and KOR in the spinal cord were experimentally compromised by the administration of opioid agonists [51,52].

Our study found expressions of the melatonin and opioid receptors MT1, MT2, MOR and KOR to be significantly higher in the cuneate nucleus and dorsal horns of burned rats treated with HBOT for one week compared with those in the untreated burned rats, indicating that HBOT alleviated burn-induced neuropathic pain by increasing the expressions of the melatonin receptors and opioid receptors via this pain transmission pathway. The cuneate nucleus, the enclosed area of the medulla oblongata, has been associated with proprioception and pain perception [53]. Inhibition of microglial activation in the cuneate nucleus has also been reported to prevent tactile allodynia in a rat model of median nerve chronic constriction injury (CCI) [54].

The upregulation of melatonin receptors has been found to accompany increases in melatonin expression and the upregulation of opioid receptors has been found to accompany increases in melatonin receptors. In this study, HBOT most likely alleviated burn-induced neuropathic pain by increasing the expression of melatonin, which led to an elevation of melatonin receptors and opioid receptors in the dorsal horn and cuneate nucleus of burned rats receiving HBOT. We did not find an upregulation of melatonin receptors and opioid receptors in burned rats receiving two weeks of HBOT, possibly because the pain response had been alleviated after one week of HBOT. The lower the nociceptive stimulation, the lower the expression of melatonin receptors and opioid receptors.

Opioid receptor expression has been found to be reduced in one model of chronic neuropathic pain and in fibromyalgia [55,56]. MOR expression has been reported by other studies to be reduced in the spinal cord and dorsal root ganglion in a rat model of nerve injury [57,58]. The results of these studies run contrary to ours. The reason for these dissimilar results may be the related differences in our models of neuropathic pain. These studies used nerve ligation; we used burns. The peripheral nerve injuries used to induce neuropathic pain could induce different pathological changes.

We did not find upregulation of MT1, MT2, MOR or KOR in burned rats that had received HBOT for two weeks, so HBOT’s dose-dependent analgesic effect likely came about via other pathways. HBOT has been found to make use of several pathways and has been related to several factors that contribute to the alleviation of acute or chronic neuropathic pain. HBOT has been found to alleviate neuropathic pain possibly by decreasing inducible nitric oxide synthase, neuronal nitric oxide synthase, inflammatory factors [59], elevating autophagy flux [60] and inhibiting the galectin-3-dependent toll-like receptor-4 pathway [12]. Further study is needed to clarify mechanisms underlying the dose-dependent analgesic effect of two weeks of HBOT.

Two pain-related neuropeptides, Substance P and CGRP, have been associated with many pain syndromes, including knee joint pain and lower back pain [38,39]. Substance P and CGRP in skin are upregulated in the skin of rats with neuroinflammation and neuropathic pain [51,52]. Substance P and CGRP regulate inflammation by binding with receptors expressed on nerves, immune cells, and epithelial cells. The immunohistochemical studies on epithelial cells we performed in this study found lower expressions of Substance P and CGRP in HBOT-treated burned rats, and rats treated for two weeks had even lower expressions of Substance P and CGRP than those treated for one week. Thus, HBOT was found to dose-dependently downregulate Substance P and CGRP. Another pain-related neuropeptide, BDNF is an important modulator within the central nervous system and spinal BDNF signaling has been found to regulate nociceptive transmission and central sensitization [61,62]. In patients with irritable bowel syndrome, increased expression of BDNF has been associated with higher abdominal pain scores [63]. In neuropathic pain, there is increased expression of BDNF in the spinal cord [64]. Of special interest, melatonin is known to exert its anti-nociceptive effect by decreasing BDNF levels [65]. Our immunohistochemical study found a decrease in BDNF in the dorsal horn of burned rats treated with HBOT compared to untreated burned rats. Considered together, our findings and those of these studies suggest that HBOT exerts its analgesic effect by increasing melatonin receptor levels which then lower BDNF levels in the central nervous system and the peripheral nervous system; however, we did not measure its expression in paw skin.

The most abundant cell type in the central nervous system is astrocyte. Numerous studies have confirmed that astrocytes play an important role in neuropathic pain [66]. They activate when there is tissue damage or peripheral nerve injury and their activation can lead to chronic pain [67]. Activated astrocytes have been associated with chemokines, neurotrophic factors, and inflammatory mediators, all known to cause or contribute to the development of thermal hyperalgesia and mechanical allodynia [68]. Some studies have experimented with downregulating the activation of astrocytes as a way of reducing neuropathic pain [67,69]. Astrocytes are also the most abundant glial cells in the brain. Glial cells are involved in normal and pathological brain function. Peripheral nerve injury signal is introduced into the central nervous system and chemical mediators are released from neurons. These chemical mediators induce the activation of glial cells, such as astrocytes and microglia [70,71,72]. Activated glial cells release chemokines and neuropeptides, which further influence neuronal activity. The interaction between neurons and glial cells plays an important role in the development of neuropathic pain [73]. GFAP is a known marker for astrocyte [74]. Our study found a decreased expression of GFAP in the dorsal horns of the HBOT-treated rats compared to the untreated rats. A lower expression of astrocyte should correspond to less neuron-glia interaction. Our results indicate that HBOT attenuated burn-induced neuropathic pain via reduction of neuron-glia interaction.

HBOT and melatonin supplementation have been used in the treatment of gastric cancer and the reduction of oxidative stress in rats [75,76]. In the current animal study of burn-induced neuropathic pain, we found HBOT to have an anti-allodynia effect in treated mice. These mice also had HBOT-induced increased levels of melatonin receptors. We believe that melatonin and HBOT might potentially be combined to treat people with burn-induced neuropathic pain.

This study has some limitations. One limitation is that we did not directly measure melatonin, thus we can only suspect the melatonin involvement based on our measurement of melatonin receptors. Another limitation is that we only measured the pain-related neuropeptide BDNF in the dorsal horn sections. Thus, we do not know how its expression was affected at the site of injury.

## 4. Materials and Methods

### 4.1. Experimental Animals

This study was approved by the Institutional Animal Care and Use Committee of Kaohsiung Medical University (IACUC Approval No.107063 1 July 2018). Thirty 5-week-old male Sprague-Dawley (SD) rats (160–180 g) were provided by Taiwan’s National Laboratory Animal Center. They were housed at Kaohsiung Medical University Animal Center where they were provided free access to water and food. The 30 rats were randomly divided into five groups (three sham groups and two full-treatment groups), six rats each: (1) Sham-burn with sham-HBOT, (2) sham-burn with HBOT group, (3) burn with sham-HBOT, and (4) burn with 1 week of HBOT or (5) burn with 2 weeks of HBOT. Once the experiments were completed, the experimental animals were anesthetized with subcutaneous injection of Zoletil (50 μg/g) and decapitated for the collection of the dorsal horn, medulla, and paw skin tissue samples. Rats of the sham-burn group were not burned on the right hind paw skin and the sham-HBOT group were treated under conditions of 1 atm (atmosphere).

### 4.2. Full-Thickness Burn Injury Procedure

To perform the burn injury, we first anesthetized the SD rats with an intraperitoneal injection of Zoletil 50 (50 μg/g; Virbac Laboratories, Carros, France). Then, a full-thickness burn injury was induced by applying the right hind paw skin to a heated metal block (75 ± 0.5 °C) for ten seconds. Silver sulfadiazine ointment was applied daily until the wounds healed and scar developed on hind paw skin approximately 3 to 4 weeks after the burn injury.

### 4.3. Dose-Dependent Hyperbaric Oxygen Treatment

Twelve burned rats were divided into two HBOT subgroups: one group that was to receive HBOT for one week, and the other for two weeks (five daily 60-min treatment sessions/week). These rats were all placed in a hyperbaric chamber (Genmall Biotechnology Co., Ltd., Taiwan) filled with 100% oxygen. The target pressure of 2.5 absolute atmospheres (ATA) was gradually achieved at a rate of 0.125 ATA/min. After the treatment, the chamber was decompressed to normal pressure at a rate of 0.125 ATA/min.

### 4.4. Assessment of Mechanical Withdrawal Threshold (MWT)

Burn injury-induced behaviors were assessed at baseline (one day before burn injury) and then followed every other week after HBOT was initiated to thirteen weeks later, at the end of the study. Mechanical pain thresholds were tested by incremental paw application of von Frey filaments (2 mm diameter) attached to a Dynamic Plantar Aesthesiometer (Ugo Basile, Comerio, Italy). Experimental animals were placed in the plastic box and allowed to acclimatize for 30 min. The mid-plantar surface of the paw was placed on a metal mesh, and the von Frey filament pressure was applied at a rate of 2.5 g/s until the animal withdrew its paw. A sudden or violent withdrawal of the paw was assumed to indicate residual intense pain. Each measurement was repeated five times.

### 4.5. Immunohistochemical (IHC) Assay

Right hind paw skin tissues were processed into 4 µm thick paraffin-embedded tissue sections. Sections were deparaffinized with xylene and rehydrated with gradient descent ethanol (100%, 95%, 70%, 50%) at room temperature. Then, hydrogen peroxide (H_2_O_2_) was added to the sections for 5 min to block endogenous peroxidase. The sections were then incubated with anti-Substance P (1:200; Sigma-Aldrich, Merck KGaA, Darmstadt, Germany) and anti-calcitonin gene-related peptide (CGRP) (1:200; Sigma-Aldrich, Merck KGaA) sequentially. Sections were incubated with the primary antibody at room temperature for 60 min. They were then washed with tris-buffered saline (TBS) and incubated with biotinylated secondary antibody Histofine^®^ Simple Stain Rat MAX PO (Nichirei Bioscience, Nichirei Corporation, Tokyo, Japan) for 1 h. Histofine^®^ Simple Stain Rat MAX PO is a labeled polymer prepared by combining amino acid polymers with peroxidase (PO) and rabbit anti-Goat IgG which are reduced to Fab fragment. Diaminobenzidine substrate was added and allowed to react for 5 min (Dako Products, Santa Clara, CA, USA). Finally, the dye, hematoxylin, was added.

Spinal cord and medulla tissues were processed into 20 µm thick optimal cutting temperature compound (OCT compound)-embedded tissue sections. After thawing at room temperature for 30 min, sections were incubated separately with anti-brain-derived neurotrophic factor (BDNF) (1:100; Novus Biologicals, LLC, Centennial, CO, USA), anti-glial fibrillary acidic protein (GFAP) (1:500; Abcam, Cambridge, UK), anti-melatonin 1A (1:100; Sigma-Aldrich, Merck KGaA), anti-melatonin 1B(1:100; Abcam), anti-MOR(1:500; Alomone Labs, Jerusalem), or anti-KOR(1:500; Alomone Labs). Sections were incubated with the primary antibody at room temperature for 60 min. Each section was then washed with tris-buffered saline (TBS) and then incubated with biotinylated secondary antibody Histofine^®^ Simple Stain Rat MAX PO (Nichirei Bioscience, Nichirei Corporation, Tokyo, Japan) for 1 h. Diaminobenzidine substrate was added to react for 5 min (Dako Products). The dye, hematoxylin, was added as the last step.

Immunohistochemical (IHC) staining scores were calculated based on the proportion of positively stained cells in the image of the tissue section. Three randomly selected fields were analyzed using the Image-Pro Plus Version 6.0 software (Media Cybernetics, Inc., Rockville, MD, USA). Human brain tissue was used as a positive control for MT1 [77] and MT2 [78]. Human skin and placenta were used as positive controls for MOR [79] and KOR [80], respectively.

### 4.6. Immunofluorescence (IF) Assay

The L3 to L4 spinal cord tissues were paraffin embedded with OCT compound, and 20 μm sections were prepared for immunostaining. After the sections were blocked in 5% normal goat serum for 1 h at room temperature, they were incubated with primary antibodies, including anti-NeuN (1:500; Millipore, Temecula, CA, USA), anti-melatonin 1A (1:100; Sigma-Aldrich, Merck KGaA), and anti-melatonin 1B (1:100; Abcam) at 4 °C overnight. The next day, the spinal cord tissue sections were incubated with secondary antibodies Alexa Fluor 488-AffiniPure Goat Anti-Mouse IgG (H+L) (1:1000; Jackson ImmunoResearch Laboratories, West Grove, PA, USA) and Goat Anti-Rabbit IgG Antibody, Cy3 conjugate (1:1000; Millipore) for 1 h at room temperature, followed by final staining with FluoroQuest™ Mounting Medium with DAPI (AAT Bioquest, Sunnyvale, CA, USA).

Stained tissue sections were analyzed using a Leica DMI6000 inverted microscope (Leica Microsystems, Inc., Buffalo Grove, IL, USA). The immunofluorescence assay staining scores were calculated based on the proportion of MT1 in NeuN positive cells and MT2 in NeuN positive cells in the merged image of the tissue section. Three randomly selected fields were analyzed using the Image-Pro Plus Version 6.0 software (Media Cybernetics, Inc., Rockville, Md.). Human brain tissue was used as positive control for MT1 [69] and MT2 [70].

### 4.7. Two-Step Real-Time Quantitative Polymerase Chain Reaction (RT-PCR) Analysis

Total RNA was extracted using a miRNeasy FFPE Kit (Qiagen, Venlo, Netherlands). RNA was measured using NanoDrop One (Thermo Scientific, Waltham, MA, USA).

The DNA sequence was evaluated using the Primer Express software. The primers were synthesized by Mission Biotech. The sequence of oligos were as follows:Rat MTNR1A-F: CACTGGCCTTCATCCTCATCTTRat MTNR1A-R: CCTGCGTTCCTGAGCTTCTTRat MTNR1B-F: TTCCTAACCATGTTCGCAGTGTRat MTNR1B-R: GAGCCATTGCCTCTGGATTG

RNA samples were reverse-transcribed for 120 min at 37 °C with a High Capacity cDNA Reverse Transcription Kit following a standard protocol suggested by the supplier (Applied Biosystems, Waltham, MA, USA). Quantitative PCR was performed under the following conditions: 10 min at 95 °C, and 40 cycles of 15 s at 95 °C, 1 min at 60 °C using 2× Power SYBR Green polymerase chain reaction (PCR) Master Mix (Applied Biosystems) and 200 nM of forward and reverse primers. Each assay was run on an Applied Biosystems 7900HT Real-Time PCR system in triplicate and fold-changes were derived using the comparative CT method. Relative quantification uses GAPDH as an endogenous control and the data of sham-burn with sham HBOT rats as a calibrator.

### 4.8. Statistical Analysis

All data were expressed as mean ± standard deviations (SD). One-way analysis of variance (ANOVA) was used to analyze all data and a Bonferroni test was performed as a posthoc test, except for in the MWT results which were analyzed with two-way ANOVA. All statistical operations were performed using SPSS 14.0 software (SPSS, Inc., Chicago, IL, USA). A *p*-value < 0.05 was considered significant.

## 5. Conclusions

In conclusion, HBOT was found to dose-dependently reduce mechanical pain, increase melatonin receptors and opioid receptors with one-week HBOT along the pain transmission pathway and decrease pain-related neuropeptides at the area of injury in burn-induced neuropathic pain in rats. In the future, we plan to further study the effects of combining melatonin and HBOT on burn-induced neuropathic pain.

## Figures and Tables

**Figure 1 ijms-20-01951-f001:**
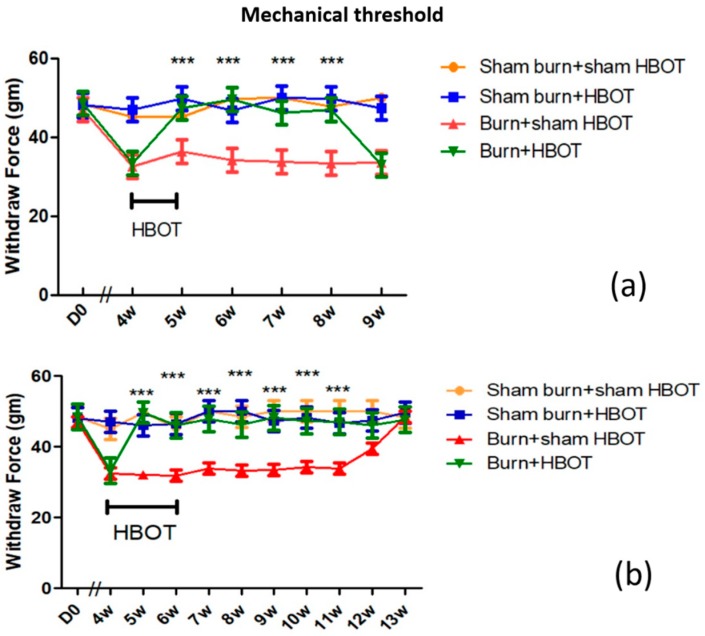
(**a**,**b**) Behavior test results of each group (*n* = 6 rats per group). (**a**) 1 week of hyperbaric oxygen treatment (HBOT), (**b**) 2 weeks of HBOT. The mechanical withdrawal threshold (MWT) increased significantly in the burned rats receiving HBOT compared to the burn with sham HBOT group. (*** *p* < 0.001; black bar represents treatment period of hyperbaric oxygen, w = weeks).

**Figure 2 ijms-20-01951-f002:**
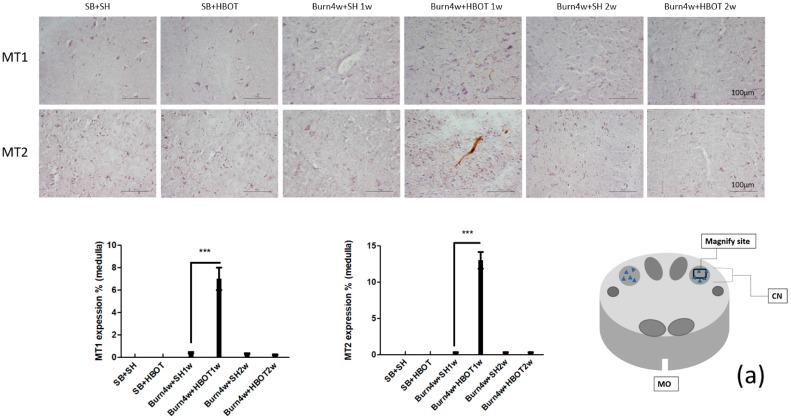
(**a**,**b**) Immunohistochemical (IHC) results of melatonin receptor 1 (MT1), melatonin receptor 2 (MT2), μ (MOR) and κ (KOR) opioid receptor in cuneate nucleus of medulla oblongata. MT1, MT2, MOR, and KOR expression significantly increased in Burn 4w with HBOT 1w compared with Burn 4w with sham HBOT 1w. The expression of MT1, MT2, MOR and KOR were not upregulated after two-weeks HBOT. (*** *p* < 0.001, original magnification: 200×, w = weeks, SB = sham burn, SH = sham HBOT, MO = medulla oblongata, CN = cuneate nucleus).

**Figure 3 ijms-20-01951-f003:**
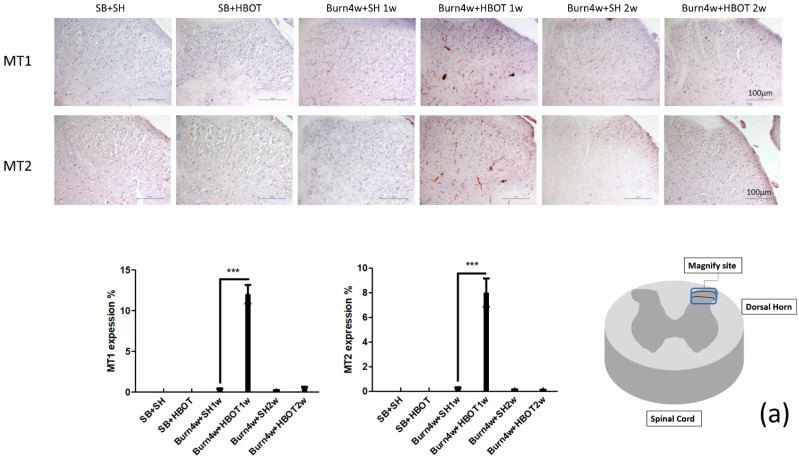
(**a**,**b**) Immunohistochemical (IHC) results of MT1, MT2, MOR, and KOR in the right dorsal horn of the spinal cord. MT1, MT2, MOR, and KOR expression significantly increased in Burn 4w with HBOT 1w compared with Burn 4w with sham HBOT 1w. The expression of MT1, MT2, MOR and KOR were not upregulated after two-weeks HBOT. (**c**) RT-PCR results of MT1 and MT2 relative quantification. The mRNA expression significantly increased in Burn 4w with HBOT 1w compared with Burn 4w with sham HBOT 1w. The mRNA expression increased in Burn 4w with HBOT 2w compared with Burn 4w with sham HBOT 2w. (*** *p* < 0.001, original magnification: 200×, w = weeks, SB = sham burn, SH = sham HBOT).

**Figure 4 ijms-20-01951-f004:**
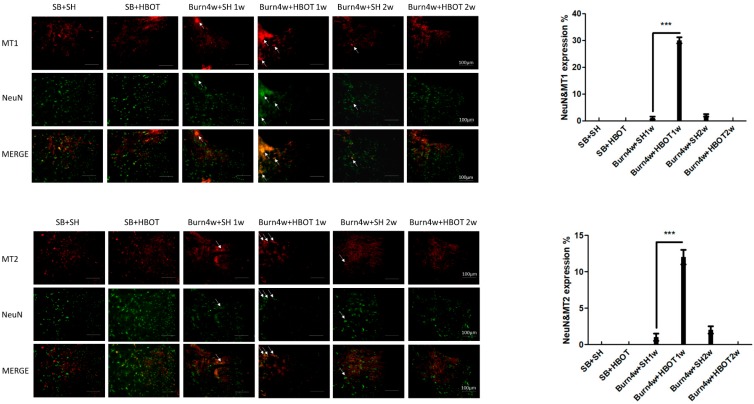
Immunofluorescence (IF) results of MT1 expression in NeuN positive cells and MT2 expression in NeuN positive cells in the right dorsal horn of the spinal cord. MT1 and MT2 expression in NeuN positive cells significantly increased in Burn 4w with HBOT 1w compared with Burn 4w with sham HBOT 1w. The expressions of MT1 and MT2 were not upregulated after two-weeks HBOT. (*** *p* < 0.001, arrows indicate stained cells, original magnification: 200×, w = weeks, SB = sham burn, SH = sham HBOT).

**Figure 5 ijms-20-01951-f005:**
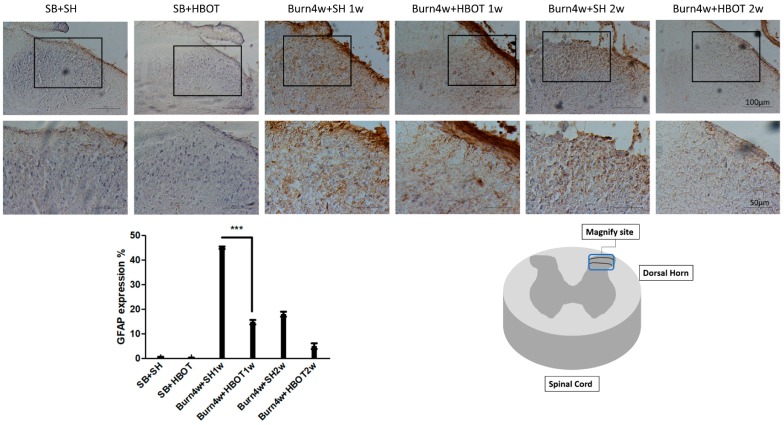
Immunohistochemical (IHC) results of glial fibrillary acidic protein (GFAP) in right dorsal horn of spinal cord. GFAP expression was significantly decreased in Burn 4w with HBOT 1w compared with Burn 4w with sham HBOT 1w. (*** *p* < 0.001, original magnification: 200×, for upper part, original magnification: 400× for lower part, w = weeks, SB = sham burn, SH = sham HBOT).

**Figure 6 ijms-20-01951-f006:**
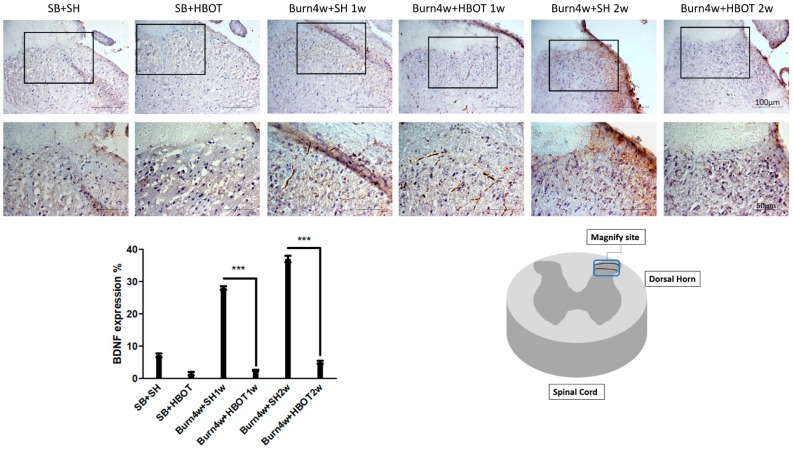
Immunohistochemical (IHC) results of brain-derived neurotrophic factor (BDNF) in right dorsal horn of spinal cord. BDNF expression was significantly decreased in Burn 4w with HBOT compared with Burn 4w with sham HBOT. (*** *p* < 0.001, original magnification: 200×, for upper part, original magnification: 400× for lower part, w = weeks, SB = sham burn, SH = sham HBOT).

**Figure 7 ijms-20-01951-f007:**
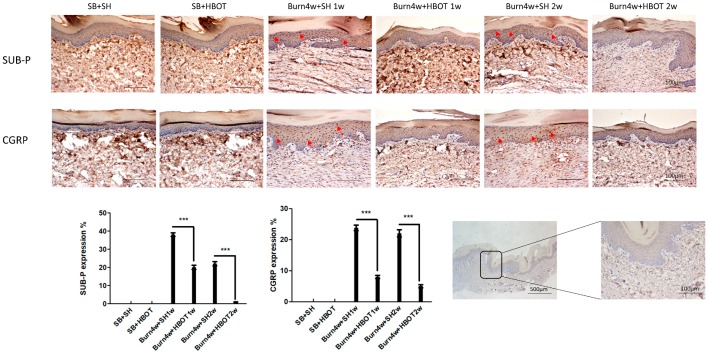
Immunohistochemical (IHC) results of Substance-P and calcitonin gene-related peptide (CGRP) in right hind paw skin. Expressions of Substance P and CGRP were significantly decreased in Burn 4w with HBOT compared with Burn 4w with sham HBOT. (*** *p* < 0.001, arrows indicate stained cells, original magnification: 200×, w = weeks, SB = sham-burn, SH = sham HBOT).

**Figure 8 ijms-20-01951-f008:**
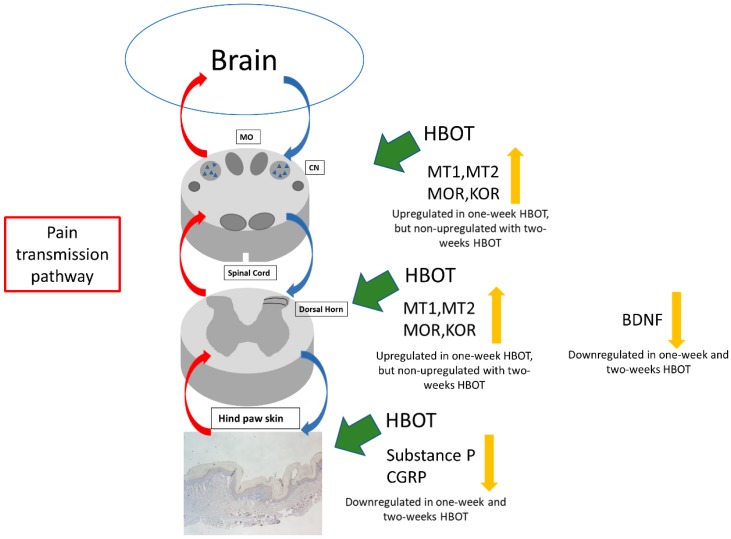
One-week HBOT upregulated MT1, MT2, MOR, and KOR expression in the cuneate nucleus of medulla oblongata and in the right dorsal horn of the spinal cord. HBOT downregulated BDNF in the right dorsal horn and Substance P and CGRP in right hind paw skin. HBOT alleviated burn-induced neuropathic pain via this pain transmission pathway (MO = medulla oblongata, CN = cuneate nucleus).

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
