# Peer review of "Dose-Dependent Effect of Hyperbaric Oxygen Treatment on Burn-Induced Neuropathic Pain in Rats"

_ijms, 2019, doi:10.3390/ijms20081951_

Reviewer 1 Report

I think that the  investigators have certainly investigated much time and effort into this  project.  Some of the things requested by the other reviewer call for  expanding the scope of this research beyond  the original project.  For other comments, I favor requesting the  authors to acknowledge deficiencies or alternative explanations in their  Discussion rather than doing additional experiments. 

For instance, I  understand the other reviewer’s comment that MWT alone is insufficient  for claiming that HBOT reduces neuropathic pain via melatonin and opioid  receptors.  However, this could have been  handled by the authors acknowledging the need for additional  verification such as antagonism by melatonin and opioid receptor  antagonists.

In my experience,  increasing the number of treatments is equivalent to increasing the  dose since hyperbaric chambers have a ceiling level of pressure that is  attainable.

Other studies  have suggested that long-term HBOT-induced pain relief may have  different underlying mechanisms for earlier and later phases of the  effect (Chung, J. Pain 9:847-53, 2010).  This may explain  some of the temporal inconsistencies pointed out by the other reviewer.

The reviewer  seems to be more knowledgeable about molecular biology techniques than I  (I only dabble in elementary molecular biology).  But I do agree with  the other reviewer that some of the figures are  very small and difficult to read. 

Author Response

Thank you for spending time reviewing my manuscript.

I think that the investigators have certainly investigated much time and effort into this project.  Some of the things requested by the other reviewer call for expanding the scope of this research beyond the original project.  For other comments, I favor requesting the authors to acknowledge deficiencies or alternative explanations in their Discussion rather than doing additional experiments. 

Answer: We have added RT-PCR in our research and alternative explanations in Discussion.

For instance, I understand the other reviewer’s comment that MWT alone is insufficient for claiming that HBOT reduces neuropathic pain via melatonin and opioid receptors.  However, this could have been handled by the authors acknowledging the need for additional verification such as antagonism by melatonin and opioid receptor antagonists.

Answer: Our Immunocytochemistry and immunofluorescence stain results also reveal that HBOT reduces neuropathic pain via melatonin and opioid receptors. We will do further study about melatonin and opioid receptor antagonists in the future.

In my experience, increasing the number of treatments is equivalent to increasing the dose since hyperbaric chambers have a ceiling level of pressure that is attainable.

Answer: Thank you for your explanation and comprehension.

Other studies have suggested that long-term HBOT-induced pain relief may have different underlying mechanisms for earlier and later phases of the effect (Chung, J. Pain 9:847-53, 2010).  This may explain some of the temporal inconsistencies pointed out by the other reviewer.

Answer: Thank you for your explanation. We have added alternative explanations about possible mechanisms involved in antiallodynia effects of HBOT in Discussion (line 243-250).

The reviewer seems to be more knowledgeable about molecular biology techniques than I (I only dabble in elementary molecular biology).  But I do agree with the other reviewer that some of the figures are very small and difficult to read. 

Answer: We have modified the size and quality of our figures.

Reviewer 2 Report

Dear Editor,

The authors propose an interesting manuscript entitled “Dose-dependent effects of Hyperbaric Oxygen Treatment on Burn-induced Neuropathic Pain via Melatonin Receptors and Opioid Receptors in Rats”. In this manuscript they have investigated the antiallodynic effect of Hyperbaric Oxygen Treatment (HBOT) in a burn-induced neuropathic model, proposing 2 weeks of HBOT as a potential therapy for the treatment of burn-induced chronic neuropathic pain. Considering the lack of therapies for the treatment of this pathological condition, this study should be carefully considered. However, the manuscript present a relevant number of issues, both in the methodology and in the analysis. Particularly, IHC results should be prudently considered and the data should be reanalysed or repeated in order to confirm their findings after a control in rat tissues. Moreover, authors do not consider/cite the relevant literature to properly frame their results.

Major corrections :

Line 69-70 (and in all the manuscript): authors should use the NC-IUPHAR approved nomenclature for the 2 melatonin receptors: MT1 (instead of MR1) and MT2 (instead of MR2)

327: “acclimatize for 10 minutes”. 10 min is a too short period for acclimation of rats. 30 min is a general accepted standard time for rats to get acclimatized.

331: “Each measurement was repeated 5 times then collect three experimental data of each rat.” It is not clear if authors chose 3 of 5 results per rat. If it is the case, how do they choose these 3 of 5? why do not consider the average after the exclusion of the outliers? Please clarify it.

Immunohistochemical (IHC) assay: please add the time of incubation for each Abs.

350: “biotinylated secondary antibody Histofine® Simple Stain Rat MAX PO”. Please clarify the source/species of the Histofine Ab(s) used for the staining. Moreover, add the source and species of primary Ab, or specify the cat# of each one.

356-358 and 374-375: Why authors have used human brain and skin tissue and placenta as positive control? And which area of the brain have been used for the control? Why do not use blocking peptides when available in the same area of investigation? Authors should clarify this choice.

Fig 4: the quality of the picture is very poor. MT1 and MT2 receptors are absent in shamBurn group. In MT1 panel the pic representing the group Burn4w+HBOT1W seem taken with a higher zoom compared to the others. Even the NeuN signal seems to be aspecific. I highly suggest to use better pictures for quantification. If authors used these pictures for the quantification, I suggest to re-run the experiments. Particularly, the signal of MT1 and MT2 have to be improved, showing as negative control the staining without the primary Abs.

Fig 5: Looking at the shown picture, it seems that the signal is stronger in shamBrun+shamHBOT compared to Burn+HBOT2W.

Fig 6 and Fig 7: Looking at the shown picture, no significant difference can be appreciated in the 6 groups. Authors should use pics with higher resolution to justify the related quantitative results.

Discussion: “regulate pain perception via its receptors MR1/MR2[39].”: a considerable amount of literature reported the role of MT2 (but not MT1) in a lot of different models of acute and chronic pain (as review please see Posa et al. 2018, and Ambriz-Tututi et al. 2009). Please reconcile all the discussion with these findings. Cite KY Ng et al 2017 instead of 40. Line 224: please clarify the term “associated”.  A large body of literature (last but not least Thompson, et al. Pain. 2018 Sep; 159(9): 1856–1866.) have demonstrated that opioid receptors, particularly MOR, are downregulated at central and peripheral level (spinal cord and dorsal root ganglion; striatum, insula, thalamus, ACC, posterior temporal and orbitofrontal cortices, and posterior midbrain) in chronic pain condition both in rodents and humans. Authors should include these results in their discussion and reconciliate the lack of downregulation of MOR in their neuropathic burn model.

277: “Astrocyte are also known as glial cells”. Astrocytes are the most abundant glia cells in the brain. Not the only ones. Please check the plural for “Astrocytes”.

393 4.8. Statistical analysis: in Fig. 1 time and treatment are the 2 considered parameters. Authors should have used a 2-way ANOVA instead of a one-way ANOVA to correctly analyze their results.

Minor corrections:

Line 68 : here authors should cite an updated review about melatonin in inflammatory and neuropathic pain: Posa et al. Targeting Melatonin MT2 Receptors: A Novel Pharmacological Avenue for Inflammatory and Neuropathic Pain. Curr Med Chem. 2018;25(32):3866-3882. Ambriz-Tututi et al. 2009.

Line 71: See comment in the Discussion about MT1 contribution in pain. Please cite a relevant review such as Posa et al. Curr Med Chem. 2018;25(32):3866-3882 or Ambriz-Tututi et al. 2009.

Line 69: here authors should cite these manuscripts: 1) Ng KY Leong MK, Liang H, Paxinos G. Melatonin receptors: distribution in mammalian brain and their respective putative functions. Brain Struct Funct. 2017 Sep;222(7):2921-2939; 2) Lacoste B. Anatomical and cellular localization of melatonin MT1 and MT2 receptors in the adult rat brain. J Pineal Res. 2015 May;58(4):397-417.

313: please add the subject of the verb “healed” or rephrase the sentence. 335

Author Response

Thank you for spending time reviewing my manuscript.

Major corrections :

Line 69-70 (and in all the manuscript): authors should use the NC-IUPHAR approved nomenclature for the 2 melatonin receptors: MT1 (instead of MR1) and MT2 (instead of MR2)

Answer: We have modified the description.

327: “acclimatize for 10 minutes”. 10 min is a too short period for acclimation of rats. 30 min is a general accepted standard time for rats to get acclimatized.

Answer: We have modified the description.

331: “Each measurement was repeated 5 times then collect three experimental data of each rat.” It is not clear if authors chose 3 of 5 results per rat. If it is the case, how do they choose these 3 of 5? why do not consider the average after the exclusion of the outliers? Please clarify it.

Answer: Because the variation is big between experimental data. Therefor, we choose 3 similar data and exclude two less relative data.

Immunohistochemical (IHC) assay: please add the time of incubation for each Abs.

Answer: We have added the description in line 336-337.

350: “biotinylated secondary antibody Histofine® Simple Stain Rat MAX PO”. Please clarify the source/species of the Histofine Ab(s) used for the staining. Moreover, add the source and species of primary Ab, or specify the cat# of each one.

Answer: We have added the description in line 339-341.

356-358 and 374-375: Why authors have used human brain and skin tissue and placenta as positive control? And which area of the brain have been used for the control? Why do not use blocking peptides when available in the same area of investigation? Authors should clarify this choice.

Answer: Positive control selected according to the datasheet of antibody. Hypothalamus tissue used for the control.

Fig 4: the quality of the picture is very poor. MT1 and MT2 receptors are absent in shamBurn group. In MT1 panel the pic representing the group Burn4w+HBOT1W seem taken with a higher zoom compared to the others. Even the NeuN signal seems to be aspecific. I highly suggest to use better pictures for quantification. If authors used these pictures for the quantification, I suggest to re-run the experiments. Particularly, the signal of MT1 and MT2 have to be improved, showing as negative control the staining without the primary Abs.

Answer: We have replaced all pictures in Fig 4.

Fig 5: Looking at the shown picture, it seems that the signal is stronger in shamBrun+shamHBOT compared to Burn+HBOT2W.

Answer: The positive-stained astrocytes is brown color. The brown color signal of Burn+HBOT2W is stronger than shamBrun+shamHBOT.

Fig 6 and Fig 7: Looking at the shown picture, no significant difference can be appreciated in the 6 groups. Authors should use pics with higher resolution to justify the related quantitative results.

Answer: We have replaced pictures of Fig 6 and Fig 7.

Discussion: “regulate pain perception via its receptors MR1/MR2[39].”: a considerable amount of literature reported the role of MT2 (but not MT1) in a lot of different models of acute and chronic pain (as review please see Posa et al. 2018, and Ambriz-Tututi et al. 2009). Please reconcile all the discussion with these findings. Cite KY Ng et al 2017 instead of 40. Line 224: please clarify the term “associated”.  A large body of literature (last but not least Thompson, et al. Pain. 2018 Sep; 159(9): 1856–1866.) have demonstrated that opioid receptors, particularly MOR, are downregulated at central and peripheral level (spinal cord and dorsal root ganglion; striatum, insula, thalamus, ACC, posterior temporal and orbitofrontal cortices, and posterior midbrain) in chronic pain condition both in rodents and humans. Authors should include these results in their discussion and reconciliate the lack of downregulation of MOR in their neuropathic burn model.

Answer: We have added discussion of MT2 in line 212-216, MOR in line 237-242. Associated used as related.

277: “Astrocyte are also known as glial cells”. Astrocytes are the most abundant glia cells in the brain. Not the only ones. Please check the plural for “Astrocytes”.

Answer: We have modified the description.

393 4.8. Statistical analysis: in Fig. 1 time and treatment are the 2 considered parameters. Authors should have used a 2-way ANOVA instead of a one-way ANOVA to correctly analyze their results.

Answer: We have modified the description.

Minor corrections:

Line 68 : here authors should cite an updated review about melatonin in inflammatory and neuropathic pain: Posa et al. Targeting Melatonin MT2 Receptors: A Novel Pharmacological Avenue for Inflammatory and Neuropathic Pain. Curr Med Chem. 2018;25(32):3866-3882. Ambriz-Tututi et al. 2009. ,1

Answer: We have added these 2 references in our manuscript (reference 18,19, line 449-452).

Line 71: See comment in the Discussion about MT1 contribution in pain. Please cite a relevant review such as Posa et al. Curr Med Chem. 2018;25(32):3866-3882 or Ambriz-Tututi et al. 2009.

Answer: We have added these 2 references in our manuscript (reference 18,19, line 449-452).

Line 69: here authors should cite these manuscripts: 1) Ng KY Leong MK, Liang H, Paxinos G. Melatonin receptors: distribution in mammalian brain and their respective putative functions. Brain Struct Funct. 2017 Sep;222(7):2921-2939; 2) Lacoste B. Anatomical and cellular localization of melatonin MT1 and MT2 receptors in the adult rat brain. J Pineal Res. 2015 May;58(4):397-417.

Answer: We have added these 2 references in our manuscript (reference 22,23, line 458-462).

313: please add the subject of the verb “healed” or rephrase the sentence. 335

Answer: We have modified the description.

This manuscript is a resubmission of an earlier submission. The following is a list of the peer review reports and author responses from that submission.

Round  1

Reviewer 1 Report

1. The effect of hyperbaric oxygen therapy in different time duration under neuropathic pain conditions has been studied in previous studies (authors’ also discussed this in the discussion section). Checking it in burn-induced neuropathic pain condition may not be novel enough for this high-quality journal. Besides, in a previous publication of the same group used different time duration (one and two weeks) oxygen therapy. The only difference in this study is that they check the effect for a longer time period.

2.    In my opinion, the findings of involvement of melatonin and opioid in attenuation of burn-induced neuropathic pain-like behavior are more interesting and novel which the authors should highlight more (in the title and during presenting the scientific message of this study). But only checking the expression of receptors using immunohistochemistry/ immunofluorescence is not scientifically strong enough for this high-quality journal.

3.    The immunohistochemistry and immunofluorescence data (receptors expression) need to be supported by checking with western blot and at mRNA level by PCR. Positive and negative controls for immunohistochemistry and immunofluorescence need to be provided.

4.    The authors’ claim that “the findings suggest that HBOT successfully alleviated burn-induced neuropathic pain via melatonin receptors and opioid receptors in the brain to site of injury pathway.. (Lines 171-172)”-------only checking MWT to claim this strong statement is not sufficient. Mechanical allodynia is only one behavior under neuropathic pain conditions.

In addition, HBOT may reduce neuropathic pain using many ways (e.g., using neuronal nitric oxide synthesis, involving microglia, reduction of many inflammatory factors etc.).

5.    It seems that expression of melatonin and opioid receptors in the cuneate nucleus and dorsal horn (Figures 2, 3) is not parallel with the attenuation of mechanical allodynia. There was no significant difference between 2 Wks after sham HBOT and 2 Wks after HBOT indicating that 2 Wks application of HBOT did not change melatonin and opioid receptors, however, mechanical allodynia attenuated (Figure 1). In addition, the immunofluorescence results (Figure 4) shows that 2 Wks HBOT treatment, even reduced the melatonin receptors in the dorsal horn of the spinal cord. Therefore, the author’ claim --“HBOT successfully alleviated burn-induced neuropathic pain via melatonin receptors and opioid receptors in the brain to site of injury pathway….” is not fully justified.

6.    The immunofluorescence pictures (Figure 4) especially Burn4wk + HBOT1wk is not in optimal quality and it seems the background is high (Figure 4). High magnification pictures are also required.

7.    Usually, substance-P and CGRP are expressed on the nerve fibers in the skin. Where these neuropeptide expressed, epithelial cells? Why they did not check in nerve fibers? Checking in the nerve fibers of the skin is more relevant to neuropathic pain.

 Better to use the term pain-related neuropeptides rather than pain relative factors in the text of the manuscript.

8.    In this research hyperbaric oxygen was applied in a single concentration/dose (2.5 ATA) and the duration of this oxygen therapy was different (one and two weeks). During drawing the conclusion of this study and in several areas of the manuscript, the authors’ use the term “dose-dependent” (although in the title, they used a different term) which can confuse the readers.

9.    The bar graphs shown in the figures are very small and difficult to see. Splitting the figures may improve the visualization.

10.    The detailed method for hyperbaric oxygen treatment need to be provided (e.g., how many sessions per day, how long per day, gradual increase or decrease of ATA, was there an accumulation of steam or CO2 etc.)

11.    In materials and methods section better to use the subtitle heading as “Assessment of mechanical withdrawal threshold (MWT)” instead of “Assessment of pain behaviors” because other pain behaviors were not checked.

Additionally, a detailed method for measurement of mechanical withdrawal threshold need to be provided (e.g., was “up and down” method, for MWT measurement used? how they minimize the effect of environmental and assessors factors? how they adapt the rats with the environment? how they reduce random errors in measurement?)

Was a scar developed on the paw skin after healing of burn [induced by heated metal block (75 ± 0.5°C)]? And did the scar formation effect on the measurement of MWT?

12.    Immunohistochemistry and immunofluorescence results are shown as percentage/ proportion. Proportion to what? How they calculate? There is no information on that.

13.    It would be interesting to discuss (in the discussion section) how (mechanism) HOBT increase melatonin and opioid receptors (mechanism).

14.    It would also better to discuss the neuron-glia cross-talk (in the discussion section) when discussing the role of astrocytes in neuropathic pain.

Author Response

Thank you for spending time reviewing my manuscript.

1. Our previous study design is to investigate the effect of hyperbaric oxygen treatment in early state of burn-injury induced neuroinflammation. This study is designed to investigate optimal HBOT sessions and the relation of melatonin receptors and opioid receptors with HBOT in neuropathic pain. We find out longer sustain time with more sessions of hyperbaric oxygen treatment. It is useful information for patients. There is no other study about the sustain time of hyperbaric oxygen treatment used in burn-induced neuropathic pain.

2. We have altered the title of this study. Because our tissue sample all fixed by formalin and all spinal cord samples used for tissue sections. Thus, we used medulla samples for PCR analysis. (4.7. Two-step Real-Time Quantitative Polymerase Chain Reaction (RT-PCR) Analysis, line 476-494)

3. Positive and negative control results of immunohistochemistry and immunofluorescence provided with word file. Because this system only can attach one file, we handed over the file to the assistant editor and forward to Reviewer.

4. We revised the description in line 291-292.

Not only MWT results, but also IHC results of pain relative neuropeptide and our previous study results reveal HBOT could alleviate neuropathic pain. (line 288-291)

5.  We have discussed the reason in Discussion line 339-342. Less pain response related to higher MWT score. MWT results of 2wks HBOT is parallel with the IHC results of melatonin receptors and opioid receptors with 2wks HBOT.

6. We have replaced Fig 4. with higher resolution pictures.

7. Thank you for your suggestions. We will change our experimental design in the future. We have replaced all pain relative factors to pain-related neuropeptides.

8. We correct all the description to dose-dependent in our manuscript and revised Materials and Methods" 4.3 Dose-dependent Hyperbaric oxygen treatment.

9. We add scale numbers above all scale bars in our immunostain figures and split the bar graph to another figure.

10.  We describe the details of hyperbaric oxygen treatment in "Materials and Methods" 4.3 Dose-dependent Hyperbaric oxygen treatment. (line 415-420).

11.     The subtitle of "4.4. Assessment of pain behavior" was modified to "Assessment of Mechanical Withdrawal Threshold. (line 421)

The details of MWT described in “4.4. Assessment of Mechanical Withdrawal Threshold”. (line 426-431) Each measurement was repeated 5 times and collect three experimental data of each rat to reduce random errors.

12. It is described in Materials and Methods " 4.5. Immunohistochemical assay" (line 456-458), "4.6. Immunofluorescence assay. (line 472-475)

13.The descriptions about the mechanism of HBOT increase expression of melatonin receptors and opioid receptors added in line 335-339.

14.The description about neuron-glia cross-talk added in line 366-376.

Reviewer 2 Report

Very well-designed and executed research with results that are consistent with the hypothesis.  My only question is in 4.3. Hyperbaric oxygen treatment.  I assume that the authors meant that the hyperbaric oxygen treatment consisted of five daily 60-minute sessions/week instead of five daily 60-second sessions/week.

Author Response

Thank you for your reminder.

We have corrected the description of "4.3 Dose-dependent hyperbaric oxygen treatment".

Round  2

Reviewer 1 Report

The manuscript is improving. However, some major problems remain.

1.        One major problem is that the conclusions are not properly/fully supported by the results. The expression of melatonin and opioid receptors was not parallel with the attenuation of mechanical allodynia. Authors wrote in the discussion section of the revised manuscript that 

“We thought that is because the pain response had been alleviated after one week of HBOT. Less nociceptive stimulation associated with less expression of melatonin receptors and opioid receptors”

This argument contradicts with their findings. In “Burn 4wk + sham HBOT 1Wk” rats, the expression of melatonin receptors and opioid receptors were low, and the rats showed mechanical allodynia. High nociceptive stimulation associated with less expression of melatonin receptors and opioid receptors was evident in those rats. A high nociceptive response was observed in those rats despite low expression of melatonin receptors and opioid receptors.

2.        The results of non-upregulation of the melatonin and opioid receptors after two weeks HBOT and the presence of extended period of attenuation of mechanical allodynia after two weeks of HBOT (compared with one week HBOT) suggest that the extended period of attenuation of mechanical allodynia may not fully due to the upregulation of these receptors (which observed after one week HBOT). There may be other mechanisms which play a role to extend the ant-allodynia effect of two weeks HBOT. The authors should write a comprehensive paragraph in the discussion section mentioning their findings of expression of melatonin and opioid receptors and ant-allodynia effect observed after two weeks HBOT. Additionally, the authors should discuss the possible mechanism of their findings (clarifying the above issues). If the authors have no proper explanation of the results, that should be mentioned for the readers and modify the conclusion.

3.        In the abstract authors wrote, “We found all HBOT treated animals to have increased expressions of melatonin receptor 1, melatonin receptor 2, μ and κ opioid receptors and decreased expressions of BDNF, Substance P, and CGRP.”

This is not the complete picture of the results. Melatonin and opioid receptors expression did not increase in 2 weeks of HBOT treated animals.

The authors should explicitly write the results, what they observed after one and two weeks of HBOT.

4.        In the abstract

Please replace “mechanically evaluated pain responses” with “mechanical paw-withdrawal threshold (MWT)”.

Please replace “four pain receptors as well as biomarkers of pain” with “expression of melatonin and opioid receptors and brain-derived neurotrophic factor (BDNF), Substance P, and calcitonin gene-related peptide (CGRP).

Please provide abbreviations.

5.        In my opinion, it is better to rewrite the conclusion in the abstract (lines 47-49) because this study does not directly check the central pain sensitization. The conclusion should be based on the proper interpretation of the findings.

In my opinion, similar to the following sentences may fit.

“Our findings suggest that increasing the duration of HBOT can reduce burn-induced mechanical allodynia for an extended period of time in rats. The upregulation of melatonin and opioid receptors observed after one week of HBOT may partly involve in attenuation of the mechanical allodynia. Downregulation of BDNF, substance P and CGRP may also contribute.” The authors should check the substance P, and CGRP in the nerves of the skin (please check the comment no 9, below) before coming to the conclusion of the last sentence)

6.        It seems that the authors want to make a conclusion of the study in Figure 8. The authors should explicitly write in the figure legend of Figure 8 that after two weeks HBOT, melatonin and opioid receptors did not upregulate. Upregulation of melatonin and opioid receptors may partly involve attenuating mechanical allodynia. Additionally, near the yellow colored arrows, the authors should show the findings of one and two weeks of HBOT with respect to the expression of melatonin and opioid receptors.

7.        In the text of the result section of the manuscript, the findings of melatonin and opioid receptors expression after two weeks HBOT should be included.

8.        In the figure legends of Figure 2, 3 and 4 the findings after two weeks HBOT should be included.

9.        Another major problem is that, In Figure 7, it seems there are many brown colored expression in the connective tissues areas, however, the authors only highlight the epithelial areas. The authors should check the expression in the connective tissues and check the expression in the nerve fibers performing double immunofluorescence staining using a marker for nerves (like PGP 9.5). This is important, because, without investigating the expression profile in the connective tissues of the skin (especially in the nerve fibers) it is difficult to claim that the expression of substance-P and CGRP in the skin reduced by HBOT (what the authors want to claim).   

10.    In the text and figure legend, the authors should write a specific location of expression of substance-P and CGRP. If it is only in the epithelial cells (after checking in the nerves), specify in the text. Provide references that show epithelial expression of these neuropeptides. Additionally,, discuss how the reduction of expression of substance-P and CGRP in the epithelial cells can reduce pain signals in the nerves under neuropathic condition.

11.    Please make the bar graphs of Figure 2, 3 and 7 similar to that of Figure 4 and 5, so that it would be visualized better.

12.    Please provide the positive and negative controls of antibodies used for immunohistochemistry and immunofluorescence as supplemental figures of the manuscript along with figure legends. Please provide the the method of doing positive and negative controls and mention which tissues were used for positive controls in the materials and method section and provide the reference to positive expression in those tissues.

13.    Result section: “2.2. HBOT enhancement of MR1, MR2, MOR and KOR in the Dorsal Horn and Cuneate Nucleus”

Please replace the subtitle with the following or similar: “The Effect of HBOT on the Expression of MR1, MR2, MOR and KOR in the Dorsal Horn and Cuneate Nucleus”

14.    If there is no statistically significant difference for the expression of astrocyte between Burn 4wk + sham HBOT 2Wk and Burn 4wk + HBOT 2Wk, please mention it in the text.

15.    Which posthoc test was performed with one way ANOVA? Please mention it in the statistical analysis section.

16.    In the starting paragraph of the discussion section, the author need to clearly state the findings of melatonin and opioid receptors expression after two weeks HBOT.

17.    “The findings suggest that HBOT should alleviate burn-induced neuropathic pain via melatonin receptors and opioid receptors in the brain to site of injury pathway, as depicted in Figure 8”.---- Please rewrite the sentence considering all findings of this study. Authors may refer to the comment no 5.

18.    Lines 302-305: “Another study, similar to ours in that it compared the analgesic effect of one and two weeks HBOT, found patients with lower limb complex regional pain syndrome treated for two weeks to have less pain, swelling, and allodynia than those receiving one week treatment [31]”.---- please check again the reference, it seems that in that study three weeks of Hyperbaric Oxygen Therapy was used.

19.    The HBOT may reduce neuropathic pain like behavior by many ways/mechanisms (e.g., modulating neuronal nitric oxide synthesis, involving microglia, reduction of many inflammatory factors etc.). The authors should discuss these possibilities in the discussion section. The authors should also discuss the mechanisms they observed in their previous studies.

Author Response

Thank you for spending time reviewing our manuscript. Please see our responses in the attached file.

Round  3

Reviewer 1 Report

1. The authors’ response to the reviewer’s major concern stated in point 1, is self-contradictory.

The authors arguing that “The expressions of melatonin receptors and opioid receptors in burned without HBOT rats are higher than the sham-burn rats. It reveals nociceptive stimulation will induce melatonin receptors and opioid receptors expression”……

In the above sentences the authors say that burn induced mechanical allodynia, not HBOT, increased the expression of melatonin and opioid receptors.

The authors continue that

“Burned rats treated with one-week HBOT show higher expressions of melatonin receptors and opioid receptors than burned rats without HBOT. It means that HBOT upregulated the expression of melatonin receptors and opioid receptors and downregulated nociceptive stimulation.”

Here the authors say that HBOT, not burn induced mechanical allodynia, increased the expression of melatonin and opioid receptors. The mechanical allodynia was attenuated due to upregulation of melatonin and opioid receptors.

The statements are self-contradictory. 

2. In response to another major concern the authors checked only PGP 9.5 (a nerve marker) expression using IHC. They did not check the substance-P and CGRP along with PGP 9.5 using double immunofluorescence staining. The reviewer concern was without investigating the expression profile of substance-P and CGRP in the connective tissues of the skin (especially in the nerve fibers) it is difficult to claim that the expression of substance-P and CGRP in the skin reduced by HBOT (what the authors want to claim). In Figure 7, it seems there are many brown colored expression of substance-P and CGRP in the connective tissues areas.

Therefore, the authors have not properly addressed the reviewer concern.

3. If we carefully check the figures for PGP 9.5 expression (provided in the response letter)

a. There are only few PGP 9.5 expressing nerve fibers in the sham burn +sham HBOT rats. Normally there should be many nerve fibers. PGP 9.5 is widely used as general marker for the peripheral nervous system.

In normal skin there should be lot of PGP9.5 expressing nerve fibers

The following paper can be checked

Hsieh, S. T., & Lin, W. M. (1999). Modulation of keratinocyte proliferation by skin innervation. Journal of investigative dermatology113(4), 579-586.

b. There was almost no PGP 9.5 expressing nerve fibers in the sham burn + HBOT rats. Why? HBOT degenerate or destroy the nerve fibers?

c. In burn 4 wk rats there were no PGP 9.5 expressing nerve fibers and the figure of burn 4 wk + sham HBOT shows a lot of expression in the epithelial tissue. Authors have not clarified why a nerve marker expressed robustly in the epithelial tissues and why there is no PGP 9.5 expressing nerve fibers.

No nerve fibers in burn 4 wk rats create a big problem. The burn 4 wk rats showed mechanical allodynia. If there are no nerve fibers in burn 4 wk rats, how the rats feel pain!!

4. Previous published papers give evidence of regeneration of Substance P, PGP 9.5 expressing nerve fibers after 2 to 4 weeks of burn injury.

a. Kishimoto, S. (1984). The regeneration of substance P-containing nerve fibers in the process of burn wound healing in the guinea pig skin. Journal of investigative dermatology83(3), 219-223.

b. Saffari, T. M., Schüttenhelm, B. N., van Neck, J. W., & Holstege, J. C. (2018). Nerve reinnervation and itch behavior in a rat burn wound model. Wound Repair and Regeneration26(1), 16-26.

5. In response to point no 10, the authors wrote in the response letter that they have discussed of the reduction of expression of substance-P and CGRP in the epithelial cells.

However, in text they only state “skin”. Skin contain epithelial and connective tissue components. They did not check the expression of substance-P and CGRP in the connective tissue components (although in Figure 7, it seems there are many brown colored expression in the connective tissues areas).

Authors also have not discussed how the reduction of expression of substance-P and CGRP in the epithelial cells can reduce pain signals in the nerves under neuropathic condition (suggested by the reviewer).